# Small-Molecule Cyclophilin Inhibitors Potently Reduce Platelet Procoagulant Activity

**DOI:** 10.3390/ijms24087163

**Published:** 2023-04-12

**Authors:** Jens Van Bael, Aline Vandenbulcke, Abdelhakim Ahmed-Belkacem, Jean-François Guichou, Jean-Michel Pawlotsky, Jelle Samyn, Arjan D. Barendrecht, Coen Maas, Simon F. De Meyer, Karen Vanhoorelbeke, Claudia Tersteeg

**Affiliations:** 1Laboratory for Thrombosis Research, KU Leuven Kulak Kortrijk Campus, 8500 Kortrijk, Belgium; jens.vanbael@kuleuven.be (J.V.B.);; 2Team Viruses, Hepatology Cancer, INSERM U955, 94000 Creteil, France; 3Centre de Biologie Structurale (CBS), INSERM U1054, CNRS UMR5048, Université de Montpellier, 34090 Montpellier, France; 4National Reference Center for Viral Hepatitis B, C and Delta, Department of Virology, Hôpital Henri Mondor, Université Paris-Est, 94000 Creteil, France; 5Diagnostic Laboratory Research, UMC Utrecht, 3584 CX Utrecht, The Netherlands

**Keywords:** Cyclophilin D, mitochondria, phosphatidylserine, platelets, procoagulant, thrombosis

## Abstract

Procoagulant platelets are associated with an increased risk for thrombosis. Procoagulant platelet formation is mediated via Cyclophilin D (CypD) mediated opening of the mitochondrial permeability transition pore. Inhibiting CypD activity could therefore be an interesting approach to limiting thrombosis. In this study, we investigated the potential of two novel, non-immunosuppressive, non-peptidic small-molecule cyclophilin inhibitors (SMCypIs) to limit thrombosis in vitro, in comparison with the cyclophilin inhibitor and immunosuppressant Cyclosporin A (CsA). Both cyclophilin inhibitors significantly decreased procoagulant platelet formation upon dual-agonist stimulation, shown by a decreased phosphatidylserine (PS) exposure, as well as a reduction in the loss of mitochondrial membrane potential. Furthermore, the SMCypIs potently reduced procoagulant platelet-dependent clotting time, as well as fibrin formation under flow, comparable to CsA. No effect was observed on agonist-induced platelet activation measured by P-selectin expression, as well as CypA-mediated integrin α_IIb_β_3_ activation. Importantly, whereas CsA increased Adenosine 5′-diphosphate (ADP)-induced platelet aggregation, this was unaffected in the presence of the SMCypIs. We here demonstrate specific cyclophilin inhibition does not affect normal platelet function, while a clear reduction in procoagulant platelets is observed. Reducing platelet procoagulant activity by inhibiting cyclophilins with SMCypIs forms a promising strategy to limit thrombosis.

## 1. Introduction

During hemostasis, platelets adhere via different surface receptors to the injured vessel wall, initiating intracellular signaling cascades that lead to platelet activation. Activated platelets, among others, secrete their granule content and activate integrin α_IIb_β_3_, which can crosslink adjacent platelets via fibrinogen, thereby inducing platelet aggregation [1]. A subpopulation of activated platelets is characterized by the expression of phosphatidylserine (PS) on the outer leaflet of the platelet membrane. These procoagulant platelets act as a scaffold for the adhesion of the tenase and prothrombinase complexes, thereby facilitating fibrin generation [2]. Procoagulant platelets are mainly observed in thrombi at the site of injury where strong platelet activation occurs via collagen and thrombin or in circulation when activated via a FcγRIIa-dependent mechanism [3,4]. In vitro, procoagulant platelets are formed by strong activation via the PAR1/4 and GPVI receptors, using dual agonist stimulation with, e.g., thrombin and collagen-related peptide (CRP).

Procoagulant platelets are required for normal hemostasis, as becomes clear from Scott Syndrome patients who suffer from a mild bleeding disorder. These patients have a null mutation in their *TMEM16F* gene, a Ca^2+^-dependent phospholipid scramblase, and hence do not expose PS on the platelet outer membrane upon activation [5]. Similarly, mice with a platelet-specific deficiency in TMEM16F develop unstable thrombi due to the absence of platelet PS exposure [6]. Procoagulant platelets also maintain vascular integrity by local initiation of coagulation during inflammation in the lung [7]. On the other hand, procoagulant platelet formation is linked to pathological thrombosis and thromboinflammation. Patients with coronary artery disease more readily form procoagulant platelets [8], and higher levels of procoagulant platelets are associated with an increased risk of recurrent stroke both in large-artery cerebrovascular disease and lacunar stroke [9,10]. Furthermore, it was shown that amplified procoagulant platelet formation increased the risk for thromboinflammation in COVID-19 patients admitted to the intensive care unit [3,11] and that procoagulant platelets are a driver of systemic thromboinflammation in mice during cerebral and gut ischemia-reperfusion (IR) injury [12]. Therefore, procoagulant platelets have been suggested as a promising therapeutic target to limit thrombosis [4]. 

An important hallmark of procoagulant platelet formation is the loss of the mitochondrial membrane potential (Δψm). The peptidyl-prolyl cis/trans isomerase Cyclophilin D (CypD), which is located within the mitochondrial matrix, acts as a key regulator of mitochondrial permeability transition pore (mPTP) opening by lowering the calcium threshold [13]. A deficiency in CypD prevents the opening of the mPTP and abolishes procoagulant platelet formation without impacting apoptosis [14]. Platelets also contain Cyclophilin A (CypA) and Cyclophilin B (CypB). While the role of platelet CypB is not clear, CypA was shown to be a Ca^2+^ regulator involved in integrin α_IIb_β_3_ bidirectional signaling [15].

The immunosuppressant Cyclosporin A (CsA) is a well-known macrocyclic cyclophilin inhibitor that potently inhibits procoagulant platelet formation via inhibition of CypD [16]. However, CsA also has several pro-thrombotic effects unrelated to cyclophilin inhibition, such as enhancing adenosine 5′-diphosphate (ADP)-induced platelet aggregation [17,18,19], possibly explaining thrombotic complications observed in patients treated with CsA [17,18,19,20,21]. This effect was implied to be independent of cyclophilin inhibition [22]. However, the lack of specific inhibitors made it, thus far, not possible to demonstrate the effect of specific cyclophilin inhibition. Recently, by means of a fragment-based drug discovery approach based on X-ray crystallography and nuclear magnetic resonance, a new family of nonpeptidic small-molecule cyclophilin inhibitors (SMCypIs) unrelated to CsA was generated [23]. These SMCypIs have potent and specific inhibitory activity against CypA, CypB, and CypD. Moreover, SMCypIs prevented mPTP opening in mitochondria from primary human and mouse hepatocytes and protected mice from liver ischemia-reperfusion injury upon binding catalytic and gatekeeper pockets of CypD [24]. 

In this study, we have tested the effect of two novel SMCypIs (F759 and F83236) on platelet function, procoagulant platelet formation and thrombus formation ex vivo and compared their effects to CsA. With this study, we aim to demonstrate the effect of specific cyclophilin inhibition to support the hypothesis that specific inhibition of platelet procoagulant activity could be a good strategy to limit thrombosis. 

## 2. Results

### 2.1. Small-Molecule Cyclophilin Inhibitors Reduce the Formation of Procoagulant Platelets

As the main effect of CsA on platelets is related to procoagulant platelet inhibition, we tested whether the SMCypIs were also able to reduce procoagulant platelet formation. First, after dual agonist stimulation using CRP-XL and thrombin, the formation of procoagulant platelets was measured via P-selectin and PS exposure, indicated by anti-CD62P and Annexin V binding, on the platelet surface. As shown in Figure 1A, dual agonist stimulation resulted in 26.9% ± 13.6 CD62P^+^/AnnexinV^+^ platelets when treated with vehicle control. We confirmed CsA was able to significantly reduce the formation of procoagulant platelets compared to vehicle control treated platelets (10.7% ± 12.3 CD62P^+^/AnnexinV^+^ platelets). Additionally, the SMCypIs significantly inhibited procoagulant platelet formation, where incubation with F759 resulted in 18.4% ± 13.1 CD62P^+^/AnnexinV^+^ platelets and F83236 in 6.5% ± 5.8 CD62P^+^/AnnexinV^+^ platelets (Figure 1A). This was accompanied by measuring the loss of the mitochondrial membrane potential upon dual agonist stimulation using tetramethylrhodamine methyl ester (TMRM) staining. In control platelets, 77.1% ± 5.8 of platelets lost their mitochondrial membrane potential in response to dual-agonist stimulation, as measured by TMRM^low^ fluorescence (Figure 1B). Both CsA, as well as SMCypIs reduced the loss of the mitochondrial membrane potential (CsA: 45.8%1 ± 5.8 TMRM^low^ platelets; F759: 47.5% ± 6.8 TMRM^low^ platelets and F83236: 35.1% ± 14.5 TMRM^low^ platelets), as shown in Figure 1B. 

Procoagulant platelets lose their membrane integrity [25], and therefore we next examined the effect of cyclophilin inhibitors on platelet morphology. Platelets were allowed to adhere to fibrinogen-coated coverslips and were then activated using dual agonist stimulation, followed by scanning electron microscopy (SEM) imaging. A clear reduction in platelets with a loss in membrane integrity (Figure 1C, indicated by arrows) was seen after preincubation of platelets with the different cyclophilin inhibitors demonstrating inhibition of procoagulant platelet formation. Procoagulant balloons were not visible using SEM as these were not able to remain adherent during sample preparation. 

Last, to assess whether inhibition of procoagulant platelet formation resulted in reduced procoagulant activity, a modified activated Partial Thromboplastin Time (aPTT) assay was used where the coagulation time is dependent on the exposure of negatively charged phospholipids [26]. This is shown by the difference in clotting time between resting platelets (NA: no agonists; 277.6 s ± 98.3) and dual-agonist activated platelets (120.0 s ± 48.1; Figure 1D). No difference in clotting time was observed between resting platelets (NA) incubated with or without cyclophilin inhibitors. Clotting times increased when dual-agonist activated platelets were incubated with CsA (186.4 s ± 55.6) and with F83236 (172.1 s ± 54.2) (Figure 1D), demonstrating a reduction in platelet procoagulant activity. Incubation with F759 resulted in an increased, however not significant, clotting time (144.5 s ± 47.0). Importantly, the clotting time in the presence of cyclophilin inhibitors and dual agonist stimulation was still faster compared to resting platelets (NA; Figure 1D), indicating that fibrin formation is not completely blocked. 

### 2.2. Cyclophilin Inhibition Has No Effect on Platelet Activation

Since CypA is involved in integrin α_IIb_β_3_ activation [15], we analyzed whether agonist-induced platelet activation was affected by specific cyclophilin inhibition. Platelets were activated with different concentrations of ADP, CRP-XL, or thrombin after incubation with the different cyclophilin inhibitors. Using flow cytometry, platelet activation was studied by detecting α-granule secretion analyzed by measuring P-selectin exposure (anti-CD62P) and by integrin α_IIb_β_3_ activation measured using the PAC-1 clone directed against activated integrin α_IIb_β_3_. Here, the Cyp inhibitors did not show any impact on the total percentage of CD62P^+^/activated integrin α_IIb_β_3_^+^ platelets (Figure 2). In addition, also the percentage of either CD62P^+^ platelets (Appendix A or the percentage of activated integrin α_IIb_β_3_^+^ (Appendix A) showed no differences. 

### 2.3. Small-Molecule Cyclophilin Inhibitors Do Not Affect Platelet Aggregation

CsA was previously shown to enhance ADP-induced platelet aggregation in vitro [17,18,19] and was therefore suggested to be responsible for the thrombotic complications observed in patients treated with CsA [17,18,20]. This effect was implied to be independent of cyclophilin inhibition [22]. However, the lack of specific cyclophilin inhibitors made it, thus far, not possible to demonstrate how specific cyclophilin inhibition modulates platelet aggregation.

We aimed to investigate whether specific cyclophilin inhibition using SMCypIs affects platelet aggregation. Therefore, we performed aggregation experiments in the presence of SMCypls or CsA using suboptimal agonist concentrations. As expected, CsA resulted in a significant enhancement of platelet aggregation upon activation with ADP compared to vehicle control (2.6 ± 1.0 fold; Figure 3A). In contrast, SMCypIs did not influence ADP-induced platelet aggregation (Figure 3A). Similarly, Horm collagen-induced aggregation in either PRP or washed platelets was not affected (Figure 3B). Lastly, no significant differences were observed in thrombin-induced platelet aggregation (Figure 3C).

### 2.4. Unaltered Primary Hemostasis but Decreased Fibrin Formation Using Small-Molecule Cyclophilin Inhibitors

Last, we investigated the effect of the cyclophilin inhibitors on platelet aggregation and fibrin formation under laminar flow. Recalcified whole blood pre-incubated with cyclophilin inhibitors was perfused over Horm collagen-coated cover slides at a shear rate of 800 sec^−1^. Platelet aggregation and fibrin formation were recorded by DIC and fluorescence microscopy (Figure 4A and Appendix A). No difference was observed in the final platelet surface coverage, indicating no effect of cyclophilin inhibition on platelet aggregation under flow (Figure 4B). This supports the results obtained using aggregometry, where cyclophilin inhibition also does not impact platelet aggregation in the presence of strong agonists, collagen and thrombin. In the vehicle control perfusions, fibrin formation started at approximately 5 min after the onset of perfusion, and the fibrin meshwork continued to increase until 9 min (full occlusion of the flow system), as shown via tPA-Alexa488 fluorescence intensity (Figure 4C,D; Δ MFI tPA-Alexa488 during 9 min: 54.9 ± 8.8). A small decrease in fibrin formation is observed when using SMCypIs F759 (Figure 4C,D; Δ MFI tPA-Alexa488: 44.2 ± 18.2) compared to the control. However, the onset of fibrin formation, as well as the maximum tPA-Alexa488 fluorescence intensity, was significantly reduced after the addition of SMCypI F83236 (Δ MFI tPA-Alexa488: 14.4 ± 10.5), as well as with CsA (Δ MFI tPA-Alexa488: 12.6 ± 7.2) (Figure 4C,D), demonstrating that inhibiting procoagulant platelets results in decreased fibrin formation. 

## 3. Discussion

Procoagulant platelet formation depends on strong agonist stimulation, resulting in CypD-mediated opening of the mPTP [14]. Ultimately, PS is exposed on the outer layer of the platelet membrane, where it acts as a scaffold for the prothrombinase complex catalyzing local fibrin formation. In the current study, we demonstrate for the first time the effect of specific cyclophilin peptidyl prolyl cis/trans isomerase activity inhibition on platelet function. First, both SMCypIs were able to reduce PS exposure and reduce the loss of the mitochondrial membrane potential after dual agonist stimulation, similar to CsA (Figure 1). Second, using a modified aPTT assay and a microfluidic flow model, a decreased fibrin formation was observed in the presence of SMCypI F83236, similar to CsA (Figure 1 and Figure 4). Additionally, last, we showed that SMCypIs did neither affect platelet activation (Figure 2) nor affect platelet aggregation (Figure 3). Therefore, the novel SMCypIs are able to potently reduce procoagulant platelet formation and platelet procoagulant activity without impacting normal platelet activation and aggregation. 

The importance of procoagulant platelets is increasingly recognized. Procoagulant platelets were shown to, among others, predict recurrent stroke in patients with a previous lacunar stroke [10], and an increased formation of procoagulant platelets is associated with adverse outcomes in patients with ischemic stroke [10]. In addition, circulating procoagulant platelets were elevated in severe intensive care unit (ICU)-admitted COVID-19 patients compared to other ICU patients [27], linked to an increased risk for thromboinflammation [3]. Therefore, inhibition of procoagulant platelets has been suggested as a possible therapeutic strategy [4]. Several targets involved in procoagulant platelet formation have been studied thus far. The absence of Aquaporin-1 in mice leads to reduced platelet membrane swelling and calcium entry, therefore diminishing PS exposure. Due to the absent PS exposure, fibrin formation was not sufficient to support thrombus stability, resulting in overall thrombus suppression without impacting bleeding time [28]. Aquaporin-1 was therefore suggested as a novel anti-procoagulant target; however, no safe and specific Aquaporin-1 inhibitors currently exist [28]. Lack of functional Orai1 in mice reduced GPVI-dependent PS exposure and protected against tissue damage during ischemic stroke without increasing the risk of intracerebral hemorrhage [29]. However, chronic inhibition of Orai1 could result in immune defects, but a short-term intervention could still be beneficial in acute ischemic diseases [30]. Pharmacological stimulation of the flippase activity of platelets was seen to reduce platelet PS exposure and fibrin formation. Yet, the authors indicated that further research is needed since the compound used appeared to be ineffective in plasma [31].

CypD inhibition has been suggested as an important target for the inhibition of IR injury in ischemic stroke [32], hepatic ischemic reperfusion injury [33,34] and myocardial infarction [35,36]. CypD is ubiquitously expressed in mammalian cells [37,38,39], and platelet CypD is an important mediator of procoagulant platelet formation. Platelet CypD plays a major role in IR injury, as mice deficient in platelet CypD were protected from thromboinflammation in severe ischemic stroke and gut ischemia [12,40]. Conflicting data have been reported on the role of platelet CypD in thrombosis; where platelet aggregation and fibrin formation were reduced in platelet-specific *CypD* deficient mice using a FeCl_3_ and laser injury model [41], a faster arterial occlusion was observed in a photochemical injury thrombosis model [42]. It was therefore hypothesized that procoagulant platelets play an important role in pathological situations where thrombin and fibrin are formed quickly [43]. In specific pathological situations, CypD inhibition could therefore be an interesting therapeutic target. Therefore, we investigated the potential of small-molecule cyclophilin inhibitors in their potential to inhibit platelet procoagulant activity. CypD shares a high degree of active site conservation across human cyclophilin isoforms, making it very difficult to develop specific CypD inhibitors [44]. Recently, a specific CypD inhibitor has been developed. However, these macrocyclic compounds were shown to be negatively charged, hindering their penetration into the cell [45]. 

CsA, a drug currently used in the context of immunosuppression after organ transplantation, has been shown to inhibit procoagulant platelets by inhibiting CypD-mediated mPTP opening. This molecule, however, is also associated with an increased risk for thrombosis in these patients [17,18,20,21], which is suggested to be mediated by an increased ADP-induced platelet aggregation [17,19,20,21]. The effect of CsA on increasing aggregation has been primarily attributed to the inhibition of calcineurin [22]. In addition, CsA is a large macromolecule and hard to adapt into derivatives; it is therefore not seen as a therapeutic option to inhibit procoagulant platelet formation. Recently, SMCypIs have been suggested as a therapeutic approach to inhibit cyclophilins in a multitude of diseases [44]. These SMCypIs were shown to potently inhibit CypA, CypB and CypD [23]. In our hands, both F759, as well as F83236 showed a reduction in platelet procoagulant activity. SMCypI F83236 was clearly more potent than F759, as F759 inhibited PS exposure to a lesser extent compared to F83236, translating into a shorter clotting time extension and less inhibition of fibrin formation under flow. Since F83236 is a more recently developed molecule, it highlights the feasibility of adapting these SMCypIs to achieve a more potent cyclophilin inhibition. 

Not only is CypD inhibited using the SMCypIs, but also CypA and CypB [23]. No role for platelet CypB has been shown thus far, but CypA was shown to affect α_IIb_β_3_ bidirectional signaling [15]. Using mice deficient in CypA, it was shown that thrombin-induced platelet aggregation was dramatically reduced, and agonist-induced integrin α_IIb_β_3_ activation was decreased [15]. Furthermore, CypA deficiency reduced Ca^2+^ mobilization from intracellular stores and Ca^2+^ entry from the extracellular space, resulting in the formation of small, unstable thrombi prone to embolization [46]. The SMCypIs, however, did not show an inhibitory effect on thrombin aggregation nor integrin α_IIb_β_3_ activation, even though the SMCypIs were shown before to potently inhibit CypA [23]. These discrepancies can possibly be explained by only partial inhibition of CypA rather than a complete genetic absence, but further studies are needed to study these differences.

In this study, we focused on the ex vivo effects of SMCypIs on human platelets. In future studies, the efficacy and safety of cyclophilin inhibition will be analyzed in murine thrombosis models where thrombin and fibrin are generated quickly after injury. Moreover, these SMCypIs also provide an interesting approach to reducing thromboinflammation induced by IR injury. In addition, assessing the bleeding risk will be important to demonstrate the safety of these compounds. A complete absence of procoagulant platelets results in mild bleeding phenotype, as seen in Scott syndrome patients lacking TMEM16F, as well as platelet-specific *Tmem16f*^−/−^ mice [6]. However, *CypD*^−/−^ mice, which can still form low numbers of procoagulant platelets via CypD-independent pathways, have a normal bleeding time [42], indicating that even small amounts of procoagulant platelets are sufficient for hemostasis. Last, platelet-specific CypD targeting might be needed to avoid effects on other cells. Targeting strategies could be used to deliver the SMCypIs to the site of platelet activation. However, in the context of IR injury, cyclophilin inhibition, broader than platelets, seems to be a viable strategy for reducing cellular necrosis [32,33,34,35,36].

In conclusion, we demonstrate for the first time that specific cyclophilin inhibition decreases platelet procoagulant activity without impacting platelet activation or aggregation, supporting the concept of inhibiting procoagulant platelets with SMCypIs as a novel strategy to reduce thrombosis.

## 4. Materials and Methods

### 4.1. Cyclophilin Inhibitors

Small molecule inhibitors of cyclophilins (SMCypIs) F759 and F83236 (chemical structure in Appendix A) were synthesized as described before [23]. Cyclosporin A (CsA) was purchased from Tocris Bioscience (cat. n° 1101, Bristol, UK).

### 4.2. Platelet Preparation

Blood from healthy volunteers was drawn into sodium citrate (9:1 blood:sodium citrate, 3.2% vol/vol). Volunteers had given written informed consent (Appendix A) in accordance with the Declaration of Helsinki and did not use platelet-inhibiting medication or anti-coagulation in the 10 days before blood withdrawal. The study was approved by the Medical Ethical Committee of KU/UZ Leuven (S64328). Blood samples were centrifuged (156× *g*, 15 min, room temperature (RT), without a break) to obtain platelet-rich plasma (PRP). Platelet-poor-plasma (PPP) was obtained by centrifugation of the remaining blood at 2000× *g* for 10 min. 

To wash platelets, acid citrate dextrose (85 mM tri-sodium citrate, 71 mM citric acid, and 111 mM D-glucose) was added (1:9 vol/vol) to PRP. Platelets were pelleted by centrifugation (330× *g*, 15 min, RT, without a break). Pelleted platelets were resuspended in *N*-2-Hydroxyethylpiperazine-*N*′-2-ethane sulphonic acid (HEPES)-buffered saline supplemented with D-Glucose (145 mM NaCl, 5 mM KCl, 0.5 Na_2_HPO_4_, 1 mM MgSO_4_, 5 mM HEPES, 5 mM D-glucose, pH 6.5) and prostacyclin (10 µg/mL). Platelets were again pelleted by centrifugation (330× *g*, 15 min, RT, without a break). Pelleted platelets were resuspended in HEPES-buffered saline supplemented with D-Glucose and with 0.2% BSA with pH 7.3. Washed platelets were allowed to rest for 30 min at RT. Where indicated, platelets were loaded with tetramethylrhodamine methyl ester (TMRM, 10 µM; cat. n° I34361, Thermo Fisher Scientific, Waltham, MA, USA) during this 30 min.

Depending on the experiment, either whole blood, PRP or washed platelets were treated for 25 min at RT, followed by 5 min at 37 °C with either 125 µM SMCypI, 4 µM CsA, or an equal volume of dimethyl sulfoxide (DMSO; vehicle control).

### 4.3. Analysis of Platelet Aggregation

For platelet aggregation experiments, PRP or washed platelets were prepared as described above. Both were used at a final concentration of 250 × 10^3^ plt/µL. Adenosine 5′-diphosphate (0.9 µM, ADP; cat. n° A5285, Sigma-Aldrich, St. Louis, MO, USA) was added to induce platelet aggregation in PRP. In washed platelets, platelet aggregation was induced by the addition of thrombin (0.1 U/mL; cat. n° HT1002a, Stago, Asnières-sur-Seine, France) or Horm collagen (2.5 µg/mL; cat. n° ABP-COL-T1, Takeda, Linz, Austria). Platelet aggregation was measured using a Chrono-Log aggregometer (Chrono-Log, Havertown, PA, USA).

### 4.4. Flow Cytometry

Washed platelets were stimulated for 10 min at 37 °C with CRP-XL (5 µg/mL) and thrombin (2 U/mL; cat. n° HT 5528PAL, Enzyme Research Laboratories, South Bend, IN, USA), or CRP-XL (0.1 or 1 µg/mL; cat. n° AB11129ML, Cambcol Laboratories, Cambridgeshire, UK), or thrombin (0.2 or 0.8 U/mL) or ADP (1 or 10 µM). After stimulation, samples were diluted with (HEPES)-buffered saline (pH 7.3) in the presence or absence of 2.5 mM CaCl_2_ and AnnexinV (APC; cat. n° 640920, Biolegend, San Diego, CA, USA). Anti-CD41 (BV421; Biolegend, San Diego, CA, USA) or anti-CD42b (Super Bright 436; cat. n° 62-0429-42, Invitrogen, Waltman, MA, USA) was used to gate platelets. AnnexinV (APC), anti-CD62P (PE; cat. n° 12-0626-80, Invitrogen, Waltman, MA, USA) and anti-activated CD41/61 (PAC-1 clone; Alexa Fluor 647; cat. n° 362806, Biolegend, San Diego, CA, USA) were used to detect respectively surface PS exposure, P-selectin exposure as a measure of α-granule release, and the open conformation of integrin α_IIb_β_3_. Loss of fluorescent signal of TMRM (TMRM^low^) stained platelets was used to detect the loss of the mitochondrial membrane potential. Fluorescence was detected using flow cytometry (BD FACSVerse, BD Biosciences, Franklin Lakes, NJ, USA).

### 4.5. Scanning Electron Microscopy 

Glass coverslips were coated with fibrinogen (100 µg/mL; cat.n° F3879, Merck-Milipore, Burlington, MA, USA) for 1.5 h at room temperature. Coverslips were then blocked with 1% BSA in HBS (500 mM HEPES, 50 mM CaCl_2_, 10 µM ZnCl_2_ 1.5 M NaCl, pH 7.4) overnight at 4°C. Following removal of the blocking reagent, washed platelets (50 × 10^3^/µL) together with Cyp inhibitors were allowed to adhere for 15 min at RT. Nonadherent platelets were removed, and adherent platelets were stimulated with thrombin (0.5 U/mL) plus CRP-XL (1 µg/mL) in HEPES buffered saline supplemented with 2.5 mM CaCl_2_. Following 15 min of stimulation at RT, agonist suspension was removed, and platelets were fixed with 2% glutaraldehyde. The coverslips were coated with 3.0 nm platina for scanning electron microscopy and imaged with a JEOL JSM-IT200 scanning electron microscope (Jeol, Akishima, Tokyo, Japan) at a 3000× magnification.

### 4.6. Modified Activated Partial Thromboplastin Time (aPTT)

Coagulation induced by procoagulant platelets was performed using a modified aPTT assay. Washed platelets were incubated with or without a combination of PAR-4 activating peptide (62.5 µM PAR4-AP; cat. n° RP11529-5, Genscript, Piscataway, NJ, USA) and CRP-XL (5 µg/mL) in the presence of 2.5 mM CaCl_2_ for 10 min. Next, platelets were incubated with PPP and ellagic acid (0.12 mM) for 2 min at 37 °C. Coagulation was initiated by the addition of 25 mM CaCl_2_. Time to clot formation was measured using a DCA-1+ coagulometer (Dutch Diagnostics, Zutphen, The Netherlands).

### 4.7. Fibrin Formation under Flow

#### 4.7.1. Preparation of Active-Site Incapacitated tPA-Alexa488 

Alexa488-conjugated active-site-inhibited tissue plasminogen activator (tPA-Alexa488) was made by incubating 35 nmol tPA (Alteplase, Boehringer Ingelheim, Ingelheim, Germany) with 350 nmol biotin-labeled PPACK for activity elimination (cat. n° BFPRCK-06 Haematologic Technologies Inc, Essex Junction, VT, USA) for 1 h at 37 degrees Celsius. tPA-PPACK-biotin conjugates were desalted with a 7K MWCO ZEBA 5 mL column using HEPES-Tyrode’s buffer (HT buffer; pH 7.4) to remove unbound biotin-labeled PPACK and subsequently mixed with alexa488-conjugated streptavidin (cat. n° S-11223, Invitrogen) in a 1:1 molar ratio (to tPA-PPACK-biotin conjugates).

#### 4.7.2. Perfusion

Perfusion experiments were performed using citrated whole blood as described by Sanrattana et al. [47]. In brief, glass coverslips (24 × 50 mm) were coated with 100 µg/mL Horm collagen in HEPES buffer (pH 7.4, 1.5h, RT) and blocked with 1% human serum albumin in HEPES buffer for 30 min. Coverslips were attached to a perfusion chamber as described [48]. Immediately before perfusion, 0.21 µg/mL anti-GP1b-VhH-AF647, 90 µM tPA-Alexa488, 6.6 mM CaCl_2_ and 3.08 mM MgCl_2_ in HEPES buffer (pH 7.4) were added to the blood. This was perfused over the Horm collagen-coated glass coverslips in a perfusion chamber at a wall shear rate of 800 s^−1^ for 9 min at RT. Platelet adhesion and fibrin formation were observed and recorded by real-time microscopy using a Zeiss Z1 observer microscope (ZEISS Microscopy, Jena, Thüringen, Germany) equipped with a 470 nm and 633 nm LED for excitation of respectively Alexa488 and Alexa647, combined with a Zeiss filter set 10 and filter set 50. Alexa488, Alexa647 and DIC images were recorded at a framerate of 2.1 frames/sec. Images were analyzed for the formation of fibrin via Alexa488 fluorescent intensity by the Image Analysis module within the ZEN2 Pro software (ZEISS Microscopy, Jena, Thüringen, Germany). Δ MFI tPA-Alexa488 was calculated by subtracting the mean tPA-Alexa488 fluorescence intensity at the start of the experiment from the MFI at 9 min.

### 4.8. Statistical Analysis

GraphPad Prism v9.0 was used (GraphPad Software, San Diego, CA, USA) for statistical analysis. One-Way ANOVA was used to compare CsA or SMCypIs to vehicles. Differences were considered significant for * *p* < 0.05, ** *p* < 0.005.

## Figures and Tables

**Figure 1 ijms-24-07163-f001:**
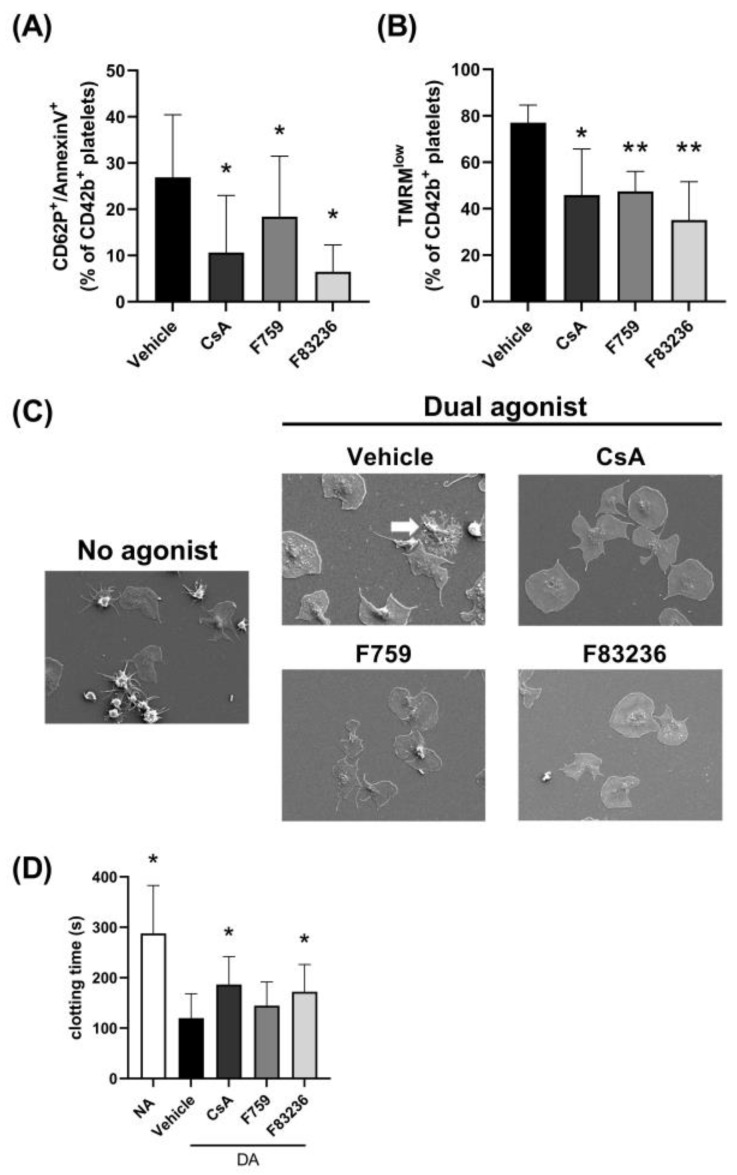
**Small-molecule cyclophilin inhibitors inhibit procoagulant platelet formation.** Washed platelets were incubated with 125 µM F759 or F83236, or 4 µM CsA. Platelets were then activated with dual agonists CRP-XL (5 µg/mL) and thrombin (2 U/mL) in the presence of 2.5 mM CaCl_2_. (**A**) During activation, platelets were labeled with anti-CD42b, anti-CD62P and AnnexinV. Using flow cytometry, CD42b^+^ platelets were gated, and the percentage of CD62P^+^/AnnexinV^+^ platelets was analyzed. (**B**) Platelets were incubated with TMRM before incubation with Cyp inhibitors. During activation, platelets were labeled with anti-CD42b. CD42b^+^/TMRM^low^ platelets were analyzed using flow cytometry. (**C**) SEM images of washed platelets. Platelets were allowed to adhere to fibrinogen-coated coverslips after incubation with Cyp inhibitors. Spread platelets are seen, as well as remnants of procoagulant platelets, which are indicated by arrows. (**D**) Washed platelets were incubated with 125 µM F759 or F83236, or 4 µM CsA. Platelets were then incubated without (NA) or with dual agonists PAR-4 (62.5 µM) and CRP-XL in the presence of 2.5 mM CaCl_2_. Next, platelets were incubated with PPP and ellagic acid before the addition of 25 mM CaCl_2_ to initiate coagulation. Data are represented as mean ± SD of 8 (panel A), 4 (panel B) or 5 (panel C) separate experiments. * *p* < 0.05, ** *p* < 0.005 compared to vehicle control by One-Way ANOVA. SMCypIs, small-molecule cyclophilin inhibitors; CRP-XL, collagen-related peptide; TMRM, tetramethylrhodamine methyl ester; Cyp, cyclophilin; CsA, Cyclosporin A; NA, no agonists; PPP, platelet-poor plasma.

**Figure 2 ijms-24-07163-f002:**
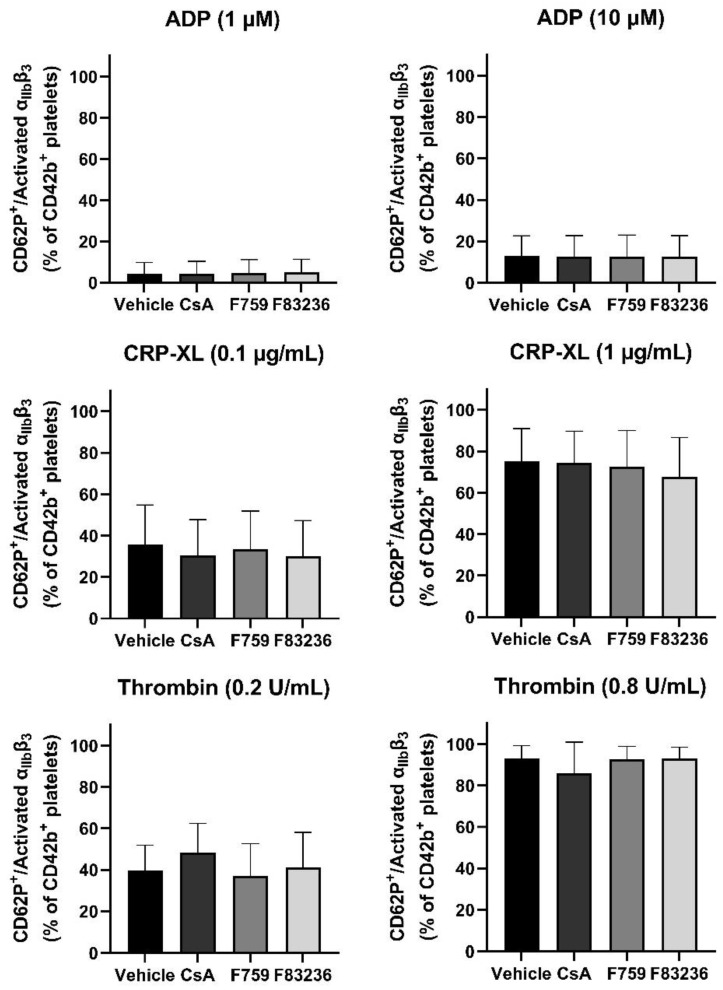
**Small-molecule cyclophilin inhibitors do not affect platelet activation.** Washed platelets were incubated with 125 µM F759 or F83236, or 4 µM CsA. Platelets were then activated with either ADP (1 or 10 µM), CRP-XL (0.1 or 1 µg/mL) or thrombin (0.2 or 0.8 U/mL). During activation, platelets were labeled with anti-CD42b, anti-CD62P and anti-activated α_IIb_β_3_ (PAC-1 clone). CD42b^+^/CD62P^+^/activated α_IIb_β_3_^+^ platelets were analyzed using flow cytometry. Data are represented as mean ± SD of 4 separate experiments. SMCypIs, small-molecule cyclophilin inhibitors; CsA, Cyclosporin A; ADP, adenosine 5′-diphosphate; CRP-XL, collagen-related peptide.

**Figure 3 ijms-24-07163-f003:**
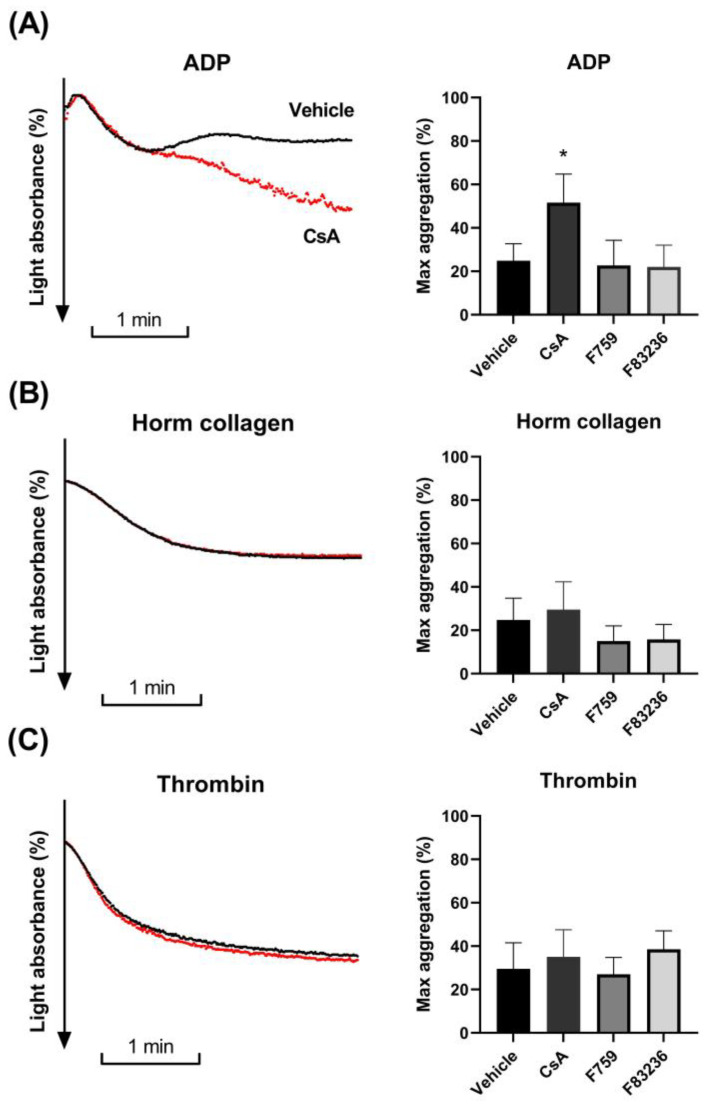
**Cyclosporin A enhances platelet aggregation, in contrast to small-molecule cyclophilin inhibitors.** Platelet Rich Plasma (PRP) or washed platelets were incubated with 125 µM SMCypls (F759 or F83236) or with 4 µM CsA. (**A**) Aggregation in PRP was induced with 0.9 µM ADP (**A**), while aggregation of washed platelets was induced by (**B**) 2.5 µg/mL Horm collagen or (**C**) 0.1 U/mL thrombin. **Left**: Aggregation curves from representative experiments upon incubation with vehicle (black) or CsA (red). **Right**: Maximal aggregation values are shown. Data are represented as mean ± SD of 4 separate experiments. * *p* < 0.05 compared to the vehicle by One-Way ANOVA. SMCypIs, small-molecule cyclophilin inhibitors; CsA, Cyclosporin A; ADP, adenosine 5′-diphosphate.

**Figure 4 ijms-24-07163-f004:**
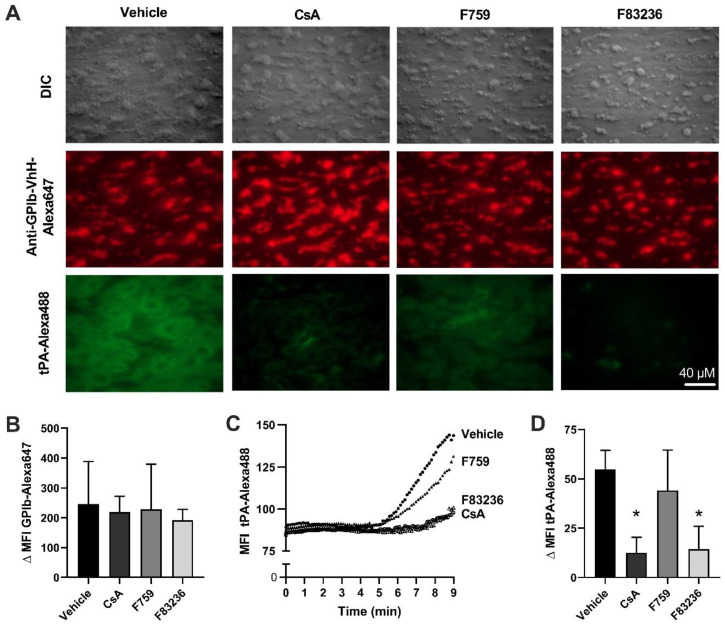
**Small-molecule cyclophilin inhibitors inhibit coagulation induced by procoagulant platelets.** (**A**) Whole blood was incubated with 125 µM F759 or F83236, 4 µM CsA before perfusion over a Horm collagen-coated surface in the presence of 6.6 mM CaCl_2_ and 3.08 mM MgCl_2_. Shown are representative images after 9 min of top row: DIC images of platelet aggregates; middle row: anti-GpIb-VhH Alexa647 indicating platelets and bottom row: tPA-Alexa488 binding to fibrin. (**B**) Quantification of Δ MFI anti-GpIb-VhH Alexa647 signal during 9 min. (**C**) MFI tPA-Alexa488 over time. (**D**) Quantification of Δ MFI tPA-Alexa488 signal during 9 min. Data represented as mean ± SD of 5–7 separate experiments. * *p* < 0.05 compared to vehicle by One-Way ANOVA. SMCypIs, small-molecule cyclophilin inhibitors; CsA, Cyclosporin A; tPA, tissue plasminogen activator.

## Data Availability

Data available on request.

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
