# Peer review of "Small-Molecule Cyclophilin Inhibitors Potently Reduce Platelet Procoagulant Activity"

_ijms, 2023, doi:10.3390/ijms24087163_

Round 1

Reviewer 1 Report

The Authors focused on the study of Small-molecule cyclophilin inhibitors potently reduce platelet procoagulant activity. I consider that the idea of this study is very interesting and has important clinical implications.

Overall, it is a well-written and comprehensive article, with clear and legible figures, interesting findings that can make significant contributions to further large studies.

In my opinion:

- The abstract presents an accurate description of this study.

- The Authors have conducted an adequate literature review.

- The references support the rationale for reporting the study.

- The management of the study is effectively described.

- Valid and reliable outcome measures were used.

- Conclusions are not appropriate and comprehensive – for better understanding by the reader, the authors should better describe the study's conclusions.

Minor suggestions:

line 60 – [9,10]

line 62 – unit [3,11]

line 78 – [17-19]

line 79 – [17-21]

line 170 – [17-19]

line 171 – [17,18,20].

line 276 – patients [17,18,20,21],

line 277 – [17,19-21]

line 396 – [17,19-21]

- please correct throughout the document

Author Response

Comments and Suggestions for Authors

 Question 1:

Conclusions are not appropriate and comprehensive – for better understanding by the reader, the authors should better describe the study's conclusions.

Response:

Thank you very much. We have adapted our conclusions and hope it is now easier to understand our conclusions.

Question 2:

line 60 – [9,10]

line 62 – unit [3,11]

line 78 – [17-19]

line 79 – [17-21]

line 170 – [17-19]

line 171 – [17,18,20].

line 276 – patients [17,18,20,21],

line 277 – [17,19-21]

line 396 – [17,19-21]

 please correct throughout the document

Response:

The references have been corrected throughout the document.

Reviewer 2 Report

The authors described that small-molecule cyclophilin inhibitors can reduce platelet procoagulant activity with ex-vivo studies. The authors explained its effect with non-peptidic small-molecule cyclophilin inhibitors and cyclosporin A and showed the potency of the reduction of platelet procoagulant activity. Even though I felt the most experiments and comments from the authors should be fine, when it comes to whole blood assay, the authors used very high concentration of samples, 125 μM F759 or F83236, 4 μM CsA before perfusion over a Horm collagen coated 213 surface, in the presence of 6.6 mM CaCl2 and 3.08 MgCl2.

Those compounds might have very high potency. Indeed, it seems like off-target might affect the inhibition of the coagulation induced by procoagulant platelets.

The authors should guarantee not to affect any artifical effect before the publication. That is my comment.

Author Response

Comments and Suggestions for Authors

 Comment:

The authors should guarantee not to affect any artificial effect before the publication. That is my comment.

 Response:

We understand the reviewer’s comment regarding possible artificial effects of the novel SMCypIs as we have used a relatively high concentration of 125 µM. This is comparable to the effect that was seen on mPTP inhibition (Panel et al., 2019, Gastroenterology) in an earlier study. SMCypIs act in vitro on the opening of the mPTP and were designed to specifically bind to cyclophilins. We have tested lower concentrations and observed a moderate inhibition of the mitochondrial permeability transition pore (mPTP) and procoagulant platelet formation. We decided to use the higher concentration to show complete inhibition without any impact on other platelet functions. Even with this high concentration, we only observe the expected effect on mPTP formation and hence procoagulant platelet formation, without any unexpected off-target effects.

Reviewer 3 Report

The manuscript entitled "Small-molecule cyclophilin inhibitors potently reduce platelet procoagulant activity" in which the authors examined investigated the potential of two novel, non-immunosuppressive, non-peptidic small-molecule cyclophilin inhibitors (SMCypIs) (F759 and F83236) to limit thrombosis in vitro, in comparison with the cyclophilin inhibitor and immunosuppressant Cyclosporin A (CsA). They demonstrated that this cyclophilin inhibition does not affect normal platelet function, while it reduces platelet procoagulant activity. The work is understandable and the topic is important. The scientific narrative is well structured and flows naturally from one idea to the next.

However, this paper suffers from some shortcomings that if modified would make the manuscript suitable for publication in International Journal of Molecular Sciences.

Shortcomings:

1-      Please discuss in brief the molecular mechanisms by which the used small-molecule cyclophilin inhibitors could reduce the thrombosis and platelet procoagulant activity.

2-      Why the authors didn’t confirm their results in vivo?

3-      How can the authors confirm the safety of the used small-molecule cyclophilin inhibitors (SMCypIs) (F759 and F83236) in their study?

4-      The authors write “In the vehicle control perfusions……………via tPA-Alexa488 fluorescence intensity (Figure 4C,D; D MFI during 9 minutes: 54.9±8.8)..”.  How was D MFI calculated?

5-      The authors write “Small molecule inhibitors of cyclophilins (SMCypIs) F759 and F83236 were synthesized as described before [23]”. Please clarify if the authors synthesize these inhibitors in their lab.

6-      Please add the full name of abbreviations in the end of each figure legend. For example SMCypIs, SEM in figure 1,….etc

7-      Are the data represented as mean+SD or mean ± SD?

8-      Please add space in P<0.05 to be P < 0.05 in each figure legend. Also put a space 26.9%±13.6 to be 26.9% ± 13.6 in line 101, 10.7%±12.3 and………....etc.

9-      Please add the catalog number of the used drug like Cyclosporin A and for the used chemicals in this study such as Adenosine 5’- diphosphate, thrombin, Horm collagen,…etc in material and methods section.

10-  Please add space between number, x, and g in (156xg, 2000xg, 330xg,……etc).

11-  Please add the template of the informant consent that was taken from subjects as an supplemental data.

12-  Please add the level of significance in statistical analysis section.

13-  Please add a section of abbreviation for easy reading.

Author Response

Question 1:

Please discuss in brief the molecular mechanisms by which the used small-molecule cyclophilin inhibitors could reduce the thrombosis and platelet procoagulant activity.

Response:

Procoagulant platelet formation is mediated via Cyclophilin D (CypD) mediated opening of the mitochondrial permeability transition pore (mPTP). Inhibiting CypD activity results in a decreased mPTP formation, which thereby limits procoagulant platelet formation. A decreased formation in procoagulant platelets will decrease the procoagulant surface that is involved in thrombin and hence fibrin formation. Inhibition of procoagulant platelet formation is therefore an interesting approach to limit thrombosis.

Question 2:

Why the authors didn’t confirm their results in vivo?

Response:

We agree that in vivo experiments will strengthen our results, however our in vitro study was performed as a proof of principle study, to demonstrate the mode of action of the SMCypIs. The half-life of the SMCypIs is currently too short to study the effect of these inhibitors in vivo. Research is currently ongoing to increase the half-life of these inhibitors, so we hope that in vivo thrombosis models will be possible in the near future.

Question 3:

How can the authors confirm the safety of the used small-molecule cyclophilin inhibitors (SMCypIs) (F759 and F83236) in their study?

Response:

The most important effect of these inhibitors regarding safety is an increased risk for bleeding. Due to the short half-life of the SMCypIs, it was unfortunately not possible to test their effect on safety in an in vivo model. However we believe our results support the safety of these inhibitors. Using both SMCypIs, fibrin formation is not completely blocked in both our modified aPTT assay as well as our flow experiments. Therefore, we believe the risk for bleeding will only be minimal. Also, no impact on other platelet functions was observed. However, future in vivo models that will be performed when the inhibitors have a prolonged half-life, will be needed to study different safety aspects.

Question 4:

The authors write “In the vehicle control perfusions……………via tPA-Alexa488 fluorescence intensity (Figure 4C,D; D MFI during 9 minutes: 54.9±8.8)..”.  How was D MFI calculated? 

Response:

D MFI tPA-Alexa488 was calculated by subtracting the mean tPA-Alexa488 fluorescence intensity at the start of the experiment from the MFI at 9 minutes. This information was added to the materials and methods section.

Question 5:

The authors write “Small molecule inhibitors of cyclophilins (SMCypIs) F759 and F83236 were synthesized as described before [23]”. Please clarify if the authors synthesize these inhibitors in their lab.

Response:

Our co-authors Abdelhakim Ahmed-Belkacem, Jean-François Guichou and Jean-Michel Pawlotsky designed and synthesized compounds F759 and F83236 in their lab. We obtained these compounds and performed the described experiments in collaboration.

Question 6:

          Please add the full name of abbreviations in the end of each figure legend. For example SMCypIs, SEM in figure 1,….etc     

Response:

Full names of abbreviations have been added to the end of each figure legend.

Question 7:

Are the data represented as mean+SD or mean ± SD?

Response:

Data are represented as mean ± SD. This has been changed in the text.

Question 8:

Please add space in P<0.05 to be P < 0.05 in each figure legend. Also put a space 26.9%±13.6 to be 26.9% ± 13.6 in line 101, 10.7%±12.3 and………....etc.   

Response:

Thank you for this comment, spaces have been added.

Question 9:

      Please add the catalog number of the used drug like Cyclosporin A and for the used chemicals in this study such as Adenosine 5’- diphosphate, thrombin, Horm collagen,…etc in material and methods section.

Response:

The catalog number of Cyclosporin A, agonists, antibodies and other relevant materials have now been added.

Question 10:

Please add space between number, x, and g in (156xg, 2000xg, 330xg,……etc).

Response:

      Spaces have been added.

Question 11:

      Please add the template of the informant consent that was taken from subjects as an supplemental data.     

Response:

The informed consent template has been added as supplemental data.

Question 12:

Please add the level of significance in statistical analysis section.

Response:

Levels of significance have been added in the statistical analysis section. Differences were considered significant for * P < 0.05, ** P < 0.005.

Question 13:

Please add a section of abbreviation for easy reading.

Response:

A list with abbreviations has been added as table 1:

Abbreviation

Full

ADP

Adenosine 5’-diphosphate

aPTT

Activated partial thromboplastin time

COVID-19

Coronavirus Disease 2019

CRP

Collagen related peptide

CsA

Cyclosporin A

CypA

Cyclophilin A

CypB

Cyclophilin D

CypD

Cyclophilin D

DIC

Differential interference contrast

DMSO

Dimethyl sulfoxide

ICU

Intensive care unit

MFI

Mean fluorescence intensity

mPTP

Mitochondrial permeability transition pore

NA

No agonists

PPP

Platelet poor plasma

PRP

Platelet rich plasma

PS

Phosphatidylserine

RT

Room temperature

SEM

Scanning electron microscopy

SMCypIs

Small-molecule cyclophilin inhibitors

TMRM

Tetramethylrhodamine methyl ester

tPA

Tissue plasminogen activator

Δψm

Loss of the mitochondrial membrane potential

Round 2

Reviewer 3 Report

The manuscript entitled "Small-molecule cyclophilin inhibitors potently reduce platelet procoagulant activity" in which the authors examined investigated the potential of two novel, non-immunosuppressive, non-peptidic small-molecule cyclophilin inhibitors (SMCypIs) (F759 and F83236) to limit thrombosis in vitro, in comparison with the cyclophilin inhibitor and immunosuppressant Cyclosporin A (CsA).

 The revised manuscript is improved compared to prior revision. My comments were adequately answered and explained by the authors. Therefore, I consider that the revised manuscript is acceptable and suitable for publication in International Journal of Molecular Sciences.